# Effect of Thermal Treatment on Corrosion Behavior of AISI 316L Stainless Steel Manufactured by Laser Powder Bed Fusion

**DOI:** 10.3390/ma15196768

**Published:** 2022-09-29

**Authors:** Francesco Andreatta, Alex Lanzutti, Reynier I. Revilla, Emanuele Vaglio, Giovanni Totis, Marco Sortino, Iris de Graeve, Lorenzo Fedrizzi

**Affiliations:** 1Polytechnic Department of Engineering and Architecture, University of Udine, via del Cotonificio 108, 33100 Udine, Italy; 2Department of Materials and Chemistry, Vrije Universiteit Brussel (VUB), Electrochemical and Surface Engineering (SURF), Pleinlaan 2, 1050 Brussels, Belgium

**Keywords:** AISI 316L stainless steel, laser powder bed fusion, heat treatment, corrosion

## Abstract

The effect of post-processing heat treatment on the corrosion behavior of AISI 316L stainless steel manufactured by laser powder bed fusion (L-PBF) is investigated in this work. Produced stainless steel was heat treated in a broad temperature range (from 200 °C to 1100 °C) in order to evaluate the electrochemical behavior and morphology of corrosion. The electrochemical behavior was investigated by potentiodynamic and galvanostatic polarization in a neutral and acidic (pH 1.8) 3.5% NaCl solution. The microstructure modification after heat treatment and the morphology of attack of corroded samples were evaluated by optical and scanning electron microscopy. The fine cellular/columnar microstructure typically observed for additive-manufactured stainless steel evolves into a fine equiaxed austenitic structure after thermal treatment at high temperatures (above 800 °C). The post-processing thermal treatment does not negatively affect the electrochemical behavior of additive-manufactured stainless steel even after prolonged heat treatment at 1100 °C for 8 h and 24 h. This indicates that the excellent barrier properties of the native oxide film are retained after heat treatment.

## 1. Introduction

Laser powder bed fusion (L-PBF) of AISI 316L stainless steel is becoming a consolidated technology for the production of near-net-shape components with complex geometry for application in different fields, including the aerospace, automotive, nuclear and biomedical sectors [1,2,3,4]. The layer-by-layer building strategy typical of the L-PBF process leads to the formation of a peculiar microstructure that is completely different from that obtained by traditional production processes [5,6,7,8]. Process parameters play a key role in the control of the microstructure and mechanical properties of L-PBF components. As a result of the very fast cooling rate of the L-PBF process, the microstructure of AISI 316L stainless steel exhibits very fine interconnected cellular or columnar austenite sub-grains with sizes in the micrometer range, confined in a macroscopic structure determined by the laser-scanning pattern and the formation of melt pools [9,10,11]. The segregation of alloying elements, including Mo, Cr and Si, is often observed at the sub-grain boundaries [10,12]. Porosity is a typical defect of additive-manufactured materials [1]. Nevertheless, the accurate control of L-PBF process parameters can guarantee the production of dense materials with a density higher than 99.8% [13,14,15]. Inclusions typically found in L-PBF AISI 316L stainless steel are fine Si- and Mn-rich oxide nanoparticles or Cr-containing silicate inclusions of a nanometric size [10,13,16,17]. MnS inclusions are usually not found in L-PBF stainless steel, although these inclusions have been reported in some cases after post-processing thermal treatment [18,19,20]. Residual stresses are unavoidable in materials produced by additive manufacturing [1,21,22]. Rapid heating rates due to high-energy input, solidification with high cooling rates in the melt pools, the melt-back effect associated with the melting of the top powder layer and the simultaneous complete or partial remelting of the underlying layers are the main sources of residual stresses in additively manufactured components [21]. This is a critical aspect since residual stresses can induce distortion in the components. Different approaches can be followed to yield a reduction in the negative effect of residual stresses in additive-manufacturing processes, including the optimization of process parameters (preheating, process planning, feedback control, laser peening) or post-processing methods (machining and thermal treatment) [21]. Post-processing thermal treatment is currently the most effective approach for the relief of residual stresses. Compressive residual stresses in the order of 250 MPa are reported for as-built AISI 316L stainless steel manufactured by L-PBF [23]. The temperature and duration of post-processing heat treatment strongly affect the magnitude of residual stresses with a nearly complete reduction for heat treatment at 1100 °C for 5 min [23]. Thermal treatment at high temperatures (above 650 °C) is associated with a marked modification of the microstructure of L-PBF AISI 316L stainless steel (annealing or recrystallization heat treatment). The annihilation of dislocations is reported for thermal treatment at 650 °C, while recrystallization takes place in the temperature range 1000–1100 °C with the disappearance of the cellular/columnar sub-grain structure and the formation of equiaxed grains [23,24,25]. The recrystallization kinetics of AISI 316L stainless steel manufactured by powder bed fusion is very slow during post-building annealing treatment at 1150 °C compared to that of its conventionally processed counterpart [26]. Recently, precipitation of MnS inclusion was reported for heat treatment above 1000 °C [20].

The corrosion performance of additively manufactured AISI 316L stainless steel has been targeted by many researchers focusing on the effect of process parameters and the main metallurgical factors, including porosity, inclusions, alloying element segregation and residual stresses [4,19]. It is generally observed that additively manufactured AISI 316L stainless steel presents higher corrosion resistance than the conventionally produced counterpart in the wrought condition [18,27,28,29,30,31,32]. The passivity of L-PBF AISI 316L stainless steel is often studied by potentiodynamic polarization, revealing a wider passive range than for the wrought counterpart. This was mainly attributed to the formation of a more protective passive oxide film on the L-PBF stainless steel than on the wrought one [14,27,31,32,33]. This aspect was recently targeted by our research group, showing that the native oxide film on L-PBF AISI 316L stainless steel displayed a consistently lower capacitance than the wrought stainless steel, indicating a different dielectric behavior despite the similarities in chemical composition and thickness of the oxide film of the investigated materials [34]. The enhanced corrosion resistance of L-PBF AISI 316L stainless steel was also associated with the formation of very fine cellular/columnar sub-grains, which are thought to promote the formation of a more protective oxide film compared to other additive-manufacturing techniques, producing more coarse microstructures or conventionally produced stainless steels [35,36]. Other authors attributed the marked passivity of additively manufactured stainless steel to the absence of deleterious MnS inclusions in L-PBF stainless steel and to the formation of nano-oxide inclusions which are too small to initiate a localized attack [18,37]. 

The effect of post-processing heat treatment on the corrosion resistance of AISI 316L stainless steel produced by additive manufacturing appears controversial according to the reported literature. Kong et al. reported that a thermal treatment at 650 °C for 30 min did not modify the passive behavior of as-built L-PBF AISI 316L stainless steel, while recrystallization annealing at 1050 °C for 30 min promoted the formation of a thicker and more protective oxide film due to the homogenization effect of the thermal treatment at a high temperature [24,25]. Similar results were reported by Zhou et al., which showed that a thermal treatment at 950 °C for 4 h produced a homogenization effect on the microstructure and distribution of alloying elements [37]. In contrast, thermal treatment at 1100 °C reduced the corrosion resistance due to the formation of a high amount of Cr(OH)_2_ and to the segregation of alloying elements after complete recrystallization [37]. Moreover, it was hypothesized that nanoscale inclusions of silicon oxide observed after heat treatment at 1100 °C could promote pit initiation [37]. Vignal et al. observed a decrease in corrosion resistance in L-PBF AISI 316L after thermal treatment at 1050 °C for 6 h, which led to the precipitation of carbides that were precursor sites for pitting [38]. Laleh et al. recently reported that pitting corrosion resistance was impaired due to the precipitation of MnS inclusions above 1000 °C [20]. In another work, Kong et al. observed a marked decrease in the pitting potential of L-PBF AISI 316L stainless steel after thermal treatment at 1050 °C and 1200 °C for 0.5, 1 and 2 h [25]. This was associated with a poorly protective thin passive film at pores that were enlarged during post-processing heat treatment. Cruz et al. investigated the effect of different stress-relieving treatments of L-PBF stainless steel, showing that compressive residual stresses tend to decrease the kinetics of oxide growth and the defect concentration in the passive film with a negative effect on repassivation behavior [23]. 

This work investigates the effect of post-processing thermal treatment in a wide temperature range (from 200 °C to 1100 °C) on AISI 316L steel manufactured by L-PBF. In particular, it focuses on microstructure modifications induced by thermal treatment and on the electrochemical behavior and morphology of the attack on post-processed 3D-printed stainless steel.

## 2. Materials and Methods

### 2.1. Materials and Post-Processing Thermal Treatment

AISI 316L stainless steel samples were manufactured by laser powder bed fusion, employing a Concept Laser M2 Cusing machine. Details about the 3D-printing process of the specimens were given in previous work from our group [27,39]. In this work, two different building planes were considered: XY samples with the main surface perpendicular to the building direction (Z axis) and XZ samples with the main surface parallel to it. Wrought stainless steel with a similar composition to that of the L-PBF specimens was employed as a reference in this study. Table 1 reports the chemical composition of L-PBF and wrought stainless steel. 

A post-processing thermal treatment was applied to L-PBF samples. The aim of thermal treatment was mainly to achieve stress relief after 3D printing, which is a necessary step for materials produced by the L-PBF process. The thermal treatment was carried out in an argon-saturated furnace for 2 h at the following temperatures: 200 °C, 400 °C, 600 °C, 800 °C, 1000 °C and 1100 °C. The L-PBF samples underwent air cooling after being removed from the furnace. In this work, L-PBF samples in the as-printed condition (without thermal treatment) are indicated by the term “as produced”. In addition to the thermal treatments described above, a set of L-PBF specimens underwent a prolonged thermal treatment at 1100 °C for 8 and 24 h in order to investigate the effect of recrystallization annealing on the microstructure and corrosion behavior of L-PBF stainless steel.

All L-PBF samples investigated in this work were ground with SiC paper in order to remove about 1 mm material from the as-printed surface. This can be considered representative of the bulk material obtained by the L-PBF process. Ground L-PBF samples and the wrought reference were mechanically polished to a 1 µm surface finish in order to enable the characterization of their microstructure and electrochemical behavior. Accurate cleaning in ethanol and rinsing in deionized water was performed prior to the characterization of the L-PBF and wrought specimens. 

### 2.2. Microstructure Characterization

The microstructure of L-PBF and wrought specimens was investigated after metallographic etching in Vilella’s reagent by means of optical and scanning electron microscopy. A Leica ICC50 HD optical microscope was employed for the investigation of the microstructure of L-PBF specimens in the as-produced condition and after thermal treatment. The microstructure was observed with a higher resolution using a ZEIS EVO 40 SEM and JSM-7610FPlus FE-SEM. 

### 2.3. Electrochemical Tests

The electrochemical behavior of L-PBF (XY and XZ surfaces) and wrought specimens was studied by means of potentiodynamic polarization measurements in a 3.5% (0.6M) NaCl solution at room temperature. The tests were performed in the electrolyte with neutral pH and with pH 1.8, which was adjusted by the addition of H_2_SO_4_ to the testing electrolyte. The polarization curves were acquired with an IPS potentiostat (Elektroniklabor Peter Schrems) using an Avesta cell (Bank Electronic—Intelligent Controls) with a three-electrode setup. The working electrode (sample) had a size of 1 cm^2^. A Ag/AgCl (3M KCl) electrode was employed as a reference. The counter electrode was a Pt wire. Samples were immersed for 15 min prior to the acquisition of polarization curves in order to yield the stabilization of the open-circuit potential. The potential was swept in the anodic direction from −0.1 V to +1.4 V vs. the open circuit potential with a scan rate of 0.2 mV/s. The acquisition of polarization curves was repeated at least three times for each specimen in order to guarantee adequate reproducibility of the results. 

In order to evaluate the corrosion morphology, L-PBF and wrought stainless steel underwent galvanostatic polarization in the same electrolytes employed for the acquisition of potentiodynamic polarization curves. A current of 1 mA/cm^2^ was applied to the samples for 1 h in order to induce a corrosion attack. This experimental approach enabled the investigation of the morphology of corrosion, ensuring that all samples underwent the same extent of attack since the same amount of charge passed in the specimens during the galvanostatic test. This was not possible in the case of the evaluation of the morphology of attack after the potentiodynamic polarization of the specimens. The surface of L-PBF and wrought specimens was investigated by optical and scanning electron microscopes after galvanostatic tests.

## 3. Results

### 3.1. Microstructure of as Produced L-PBF AISI 316L Stainless Steel

Figure 1 shows the microstructure of AISI 316L stainless steel in the wrought condition and after 3D printing by L-PBF. As can be seen in panel (a) of Figure 1, the wrought stainless steel displayed an austenitic structure with relatively large equiaxed grains. The L-PBF material revealed the laser-scan pattern in the XY and XZ planes (Figure 1b,c). In particular, overlapping melt pools with a depth in the order 50–100 μm can be recognized in panel (c). SEM observations at a higher magnification confirmed the austenitic structure of the wrought stainless steel (Figure 2a), while it highlighted the typical cellular/columnar structure in the L-PBF stainless steel (Figure 2b). The microstructure of L-PBF samples consisted of columnar austenite and decomposed ferrite located at austenite boundaries for both printing directions (XY and XZ surfaces), as already discussed in another work [39]. Moreover, the porosity content of the L-PBF samples was in the order 0.1–0.2% [27].

### 3.2. Microstructure of L-PBF AISI 316L Stainless Steel after Thermal Treatment

#### 3.2.1. XY Surface

Figure 3 displays the microstructure on the XY building plane of L-PBF AISI 316L stainless steel after thermal treatment for 2h at temperatures ranging between 200 °C and 1100 °C. The laser-scan pattern observed after printing remained clearly visible for temperatures up to 400 °C (Figure 3b,c); meanwhile, at higher temperatures, there was a progressive evolution of the microstructure towards a fine austenitic structure, resembling that observed for the wrought material but with a significantly lower grain size (Figure 3d–f).

SEM micrographs of the L-PBF stainless steel (XY building plane) are visible in Figure 4. The fine cellular/columnar microstructure typically observed for L-PBF samples (Figure 4a–c) was not clearly recognizable for thermal treatment above 800 °C (Figure 4d–f). Partial solubilization of ferrite most likely initiates above 600 °C and is completed for thermal treatment at higher temperatures. Moreover, it can be seen that the austenite structure tends to rearrange the formation equiaxed grains for thermal treatment at 800, 1000 and 1100 °C. No deleterious phases (sigma phase or precipitates) were detected during the analysis of heat-treated L-PBF stainless steel.

#### 3.2.2. XZ Surface

Figure 5 displays the microstructure on the XZ building plane of L-PBF AISI 316L stainless steel after thermal treatment for 2 h at temperatures ranging between 200 °C and 1100 °C. The melt pools can be clearly identified for thermal treatment up to 600 °C (Figure 5a–c). The thermal treatment above 800 °C confirmed the transition towards a fine austenitic structure that appeared complete for thermal treatment above 1000 °C (Figure 5d–f). SEM micrographs of the L-PBF stainless steel (XZ building plane) (Figure 6) showed that the cellular/columnar structure was no longer visible for thermal treatments above 800 °C, confirming the trend observed for the XY surface. The formation of equiaxed austenitic grains was also visible in the XZ surface for thermal treatment above 800 °C (Figure 6d–f).

### 3.3. Electrochemical Behaviour of L-PBF AISI 316L Stainless Steel after Thermal Treatment

#### 3.3.1. Potentiodynamic Polarization Curves

Figure 7 displays potentiodynamic polarization curves in a neutral 3.5% NaCl solution of AISI 316L stainless steel in the wrought condition and after 3D printing by L-PBF (as-produced condition). The polarization curves of L-PBF specimens subjected to thermal treatment for 2 h at temperatures between 200 °C and 1100 °C are also included in the figure.

The curve of the wrought stainless steel displayed passive behavior with a breakdown at +0.6 V vs. AgAgCl [3M KCl]. The breakdown was anticipated by a rather marked instability of the current density, most likely associated with metastable pitting. The curve for the L-PBF stainless steel in the as-produced condition (without thermal treatment) exhibited an extended passive range with a marked increase in current density only above +1.2 V vs. AgAgCl [3M KCl]. The increase in current density was very sharp for the wrought material at the breakdown potential, while this was more progressive for the as-produced L-PBF sample. Polarization curves of the heat-treated L-PBF samples displayed the same behavior of the material in the as-produced condition, with a large passive range. Only the sample heat treated at 1100 °C displayed some extent of metastable pitting, while samples heat treated at lower temperatures did not show metastable pitting. All the samples presented very similar corrosion current density values (about 10^−6^ Acm^−2^).

Figure 8 shows the polarization curves acquired in a 3.5% NaCl solution with pH 1.8. The wrought stainless steel exhibited a marked metastable pitting in the passive range and the breakdown occurred at about +0.4 V vs. AgAgCl [3M KCl]. This is about 200 mV more negative than the breakdown observed in the neutral solution (Figure 7). The L-PBF stainless steel in the as-produced condition exhibited a more extended passive range than its wrought counterpart, as well as in the electrolyte with pH 1.8. The anodic current density displayed a progressive increase at about +0.8 V vs. AgAgCl [3M KCl]. All the samples presented very similar corrosion current density values in a 3.5% NaCl solution with pH 1.8 (about 10^−6^ Acm^−2^). The heat-treated L-PBF samples also displayed the same behavior of the as-printed material in acid conditions, confirming the trend observed in Figure 7. 

#### 3.3.2. Morphology of Attack after Galvanostatic Polarization

The morphology of attack after galvanostatic polarization in a 3.5% NaCl solution in a neutral condition and with pH 1.8 is shown in Figure 9 for L-PBF 316L stainless steel heat treated for 2 h at 400 °C and 1100 °C. The sample heat treated at 400 °C exhibited few shallow pits with a size below 100 µm after galvanostatic polarization in a neutral 3.5% NaCl solution. As can be seen in the example in panel (a) of Figure 9, the corrosion attack revealed that the structure of the laser-scanning pattern and the typical cellular/columnar microstructure was still recognizable inside the pit. The sample heat treated at 1100 °C showed few pits with limited depth, as in the case of the sample heat treated at a lower temperature. The morphology of the pits (Figure 9b) recalled features related to the laser-scanning pattern, although the cellular/columnar microstructure was not coherently visible at the bottom of the pit with the microstructure modification induced by thermal treatment, which is shown in Figure 4 and Figure 6.

The morphology of heat-treated L-PBF stainless steel after galvanostatic polarization in the electrolyte with pH 1.8 displayed a more severe attack compared to measurements in the neutral solution. The attack tended to spread on the sample surface rather than produce deep pits. An example is shown in panel (c) of Figure 9 for the L-PBF stainless steel heat treated at 400 °C. The typical microstructure resulting from the L-PBF process remained visible at the bottom of the pit, although the attack was more pronounced in the electrolyte with pH 1.8 (Figure 9c) than in the neutral solution (Figure 9a). The L-PBF sample heat treated at 1100 °C (Figure 9d) displayed pits with a similar morphology to that observed in the neutral solution. In this case, the attack was also more marked in the electrolyte with pH 1.8.

### 3.4. Effect of Prolonged Thermal Treatment of L-PBF AISI 316L Stainless Steel

#### 3.4.1. Microstructure

In order to further investigate the effect of thermal treatment, the L-PBF-XZ 316L stainless steel was subjected to thermal treatment at 1100 °C for 8 h and 24 h in order to promote a complete evolution of the microstructure towards an austenitic structure similar to that of the wrought reference material. The SEM micrographs of the L-PBF stainless steel after thermal treatment for 8 h (Figure 10a) displayed a fine austenitic structure, with smaller grains than that of the wrought material (panel (a) of Figure 4 and Figure 6). The size of the austenitic grains also remained smaller than that of the wrought material after thermal treatment for 24 h (Figure 10b).

#### 3.4.2. Potentiodynamic Polarization Curves and Morphology of Attack

Figure 11 shows the potentiodynamic polarization curves acquired for L-PBF stainless steel samples heat treated at 1100 °C for 2 h, 8 h and 24 h. Curves for the wrought counterpart and L-PBF sample in the as-produced condition are also included in the figure. The curves after thermal treatment for 8 h and 24 h were very similar to those for the stainless-steel samples in the as-produced condition and after 2 h heat treatment, with the exception that a marked metastable pitting could be observed in the passive range. Nevertheless, the existence of a more extended passive range compared to the wrought material was also confirmed after prolonged thermal treatment.

The morphology of attack after galvanostatic polarization in a neutral 3.5% NaCl solution was also investigated after the prolonged thermal treatment of the L-PBF stainless steel, revealing a similar type of attack as for samples heat treated at 1100 °C for 2 h (Figure 9). Figure 12a,b display some examples of shallow pits that can be found on the surface of the L-PBF stainless steel after thermal treatment for 8 h and 24 h. The pit visible in panel (a) of Figure 12 was rather large, with a size of approximatively 500 µm, but it was very shallow. A similar morphology was visible after 24 h heat treatment (Figure 12b). As seen for heat treatments of 2 h, the cellular/columnar structure of the material was no longer visible inside the pits after prolonged thermal treatment.

## 4. Discussion

The printing parameters of 316L stainless steel by the L-PBF process led to the formation of a peculiar microstructure that was completely different to that of the wrought counterpart (Figure 1 and Figure 2). This was investigated in detail for the as-printed material in a previous publication [27]. The microstructure revealed the laser-scanning pattern in the XY plane (perpendicular to the building direction) and overlapping melt pools in the XZ plane (parallel to the building direction). This is superimposed to very fine cellular/columnar structures that exhibit the same morphology on both XY and XZ planes, rendering the different building planes indistinguishable at high magnification (Figure 2B). The overall quality of the as-produced material was very high since the porosity was very low (below 0.2%). Moreover, no inclusions regarding size in the micrometer range could be detected in the L-PBF stainless steel. In particular, no deleterious MnS inclusion could be detected in the L-PBF stainless steel and the wrought materials employed as a reference in this study. This is probably associated with the very low S content of materials employed in this study (Table 1) and the inherently high solidification rate during additive manufacturing, which could hinder the formation of MnS inclusions as suggested by Chao et al. [18].

Post-processing thermal treatment had a marked impact on the microstructure of L-PBF 316L stainless steel. There was a clear transition towards a fine austenitic structure, resembling that of the wrought material initiates, for thermal treatments above 800 °C (Figure 3 and Figure 5). The fine cellular/columnar microstructure typically observed for L-PBF samples (Figure 2b) was no longer visible for thermal treatment above 800 °C (Figure 4 and Figure 6). Remarkably, the austenitic grain size of the L-PBF stainless steel remained smaller than that of the wrought counterpart, even after prolonged thermal treatment at 1100 °C (Figure 10). A possible explanation for this finding is that the recrystallization kinetics of 316L stainless steel produced by laser powder bed fusion is very slow during solution annealing at 1150 °C [26].

Our previous work clearly highlighted that L-PBF 316L stainless steel in the as-produced condition exhibited higher corrosion resistance than the wrought counterpart [27,34]. This is confirmed by the potentiodynamic polarization curves in Figure 7 and Figure 8. We believe that this is associated with the better protective properties of the oxide film for the L-PBF stainless steel, despite the fact that the chemical composition and thickness of the native oxide film is similar for L-PBF and wrought specimens [27,34]. Indeed, we could show that the dielectric properties were different for L-PBF and wrought stainless steel. The high stability of the native oxide film in L-PBF stainless steel is probably promoted by the very fine cellular/columnar structure. Moreover, micro-segregation effects render the borders of these structures more corrosion resistant than their interior.

The results presented in this work clearly show that the post-processing thermal treatment does not negatively affect the electrochemical behavior of L-PBF stainless steel (Figure 7 and Figure 8). This behavior was observed even after thermal treatment at 1100 °C for 8 h and 24 h (Figure 11). This clearly suggests that the protective properties of the native oxide film on L-PBF stainless steel are not significantly modified by the thermal treatments considered in our work, despite the marked modification of the microstructure observed for thermal treatments above 800 °C. An improvement in the corrosion resistance of 316L stainless steel after thermal treatment was already reported in the literature [24,37]. Zhou et al. showed that a thermal treatment at 550 °C promoting the annihilation of dislocations and the elimination of residual stresses had a marginal effect on corrosion behavior, which is in line with our findings [37]. Moreover, they reported that thermal treatment at 950 °C promoted the elimination of sub-grain boundaries, which was associated with a partial and incomplete homogenization, making the distribution of the alloying elements more uniform than in the as-printed material [37]. The results reported in the literature regarding the corrosion behavior of 316L stainless steel heat treated at high temperatures (above 1000 °C) remain rather controversial. In some cases, it is reported that corrosion behavior might be impaired by the precipitation of deleterious MnS inclusions, which is reported for solution annealing above 1000 °C [20]. Moreover, Vignal et al. reported that the corrosion resistance of the passive oxide film can be reduced after thermal treatment at 1050 °C by the precipitation of mixed oxide particles that act as precursor sites for pitting [38]. In our work, the protective properties of the oxide film were retained for thermal treatments in the range 800–1000 °C. It is likely that the partial or even complete annihilation of the cellular/columnar structures results in a homogeneous distribution of alloying elements, retaining the protective properties of the native oxide film. Moreover, the absence of dangerous inclusions or large pores in the L-PBF stainless steel employed in our work might help to maintain the protective properties after thermal treatment at high temperatures. This is further supported by the electrochemical behavior observed after prolonged thermal treatment, for which it can be expected that complete recrystallization and homogenization is achieved. 

The morphology of attack of the as-printed L-PBF and wrought stainless steel was previously discussed in another work [27]. It was shown that, after galvanostatic polarization, the wrought stainless steel exhibits a large number of deep and large pits, while the L-PBF sample in the as-printed condition displays few pits with limited depth. Few and shallow pits are also visible after galvanostatic polarization of the L-PBF stainless steel heat treated at 1100 °C (Figure 9). In the case of the L-PBF stainless steel in the as-printed condition, the large shallow pits reveal the structure of the laser-scanning pattern highlighting the cellular/columnar structure resulting from the L-PBF process. This could be linked to the high-corrosion resistance of the borders of the cells/columns, forming a 3D-interconnected network that provides a barrier for pitting propagation inside the L-PBF material [34]. This barrier effect seems to be retained after thermal treatment, as confirmed by the morphology of attack after thermal treatment at 400 °C (Figure 9a,c). The morphology of attack after thermal treatment at 1100 °C (Figure 9b,d) is similar to that observed for heat treatment at lower temperatures, even after prolonged heat treatment (Figure 12). This suggests that partial or even complete annihilation of the cellular/columnar structures typical of as-produced L-PBF material does not affect the protective properties of the oxide film. Moreover, these protective properties could also be related to the existence of a very fine austenitic structure after prolonged thermal treatment. The high-corrosion resistance and limited attack displayed by post-processing thermal treatments considered in this work imply that a stress-relief treatment could be performed without impairing corrosion behavior, provided that the quality of the as-printed L-PBF stainless steel is adequate. This expands the applicability of 3D-printed stainless-steel components since post-processing heat treatments can be performed to eliminate residual stresses without affecting corrosion resistance.

## 5. Conclusions

The modification of the microstructure and electrochemical behavior induced by the heat treatment of AISI 316 stainless steel produced by L-PBF was investigated in this work. The main conclusions of this work are as follows:The laser-scanning pattern typically observed for the XY surface of AISI 316L stainless steel manufactured by L-PBF progressively evolves into an austenitic structure with fine grains increasing the heat-treatment temperature. A similar modification was observed for the complex overlapping melt pools visible in the XZ surface.The modification of the microstructure due to thermal treatment does not affect the electrochemical behavior of the L-PBF stainless steel, which displays the same passive behavior of the material in the as-printed condition for heat treatments up to 1100 °C.Post-processing heat treatments (stress-relief thermal treatment or recrystallization annealing) can be performed without significantly affecting the high-corrosion resistance of the as-produced L-PBF stainless steel in the temperature range studied in this work. This expands the applicability of 3D-printed stainless-steel components.

## Figures and Tables

**Figure 1 materials-15-06768-f001:**
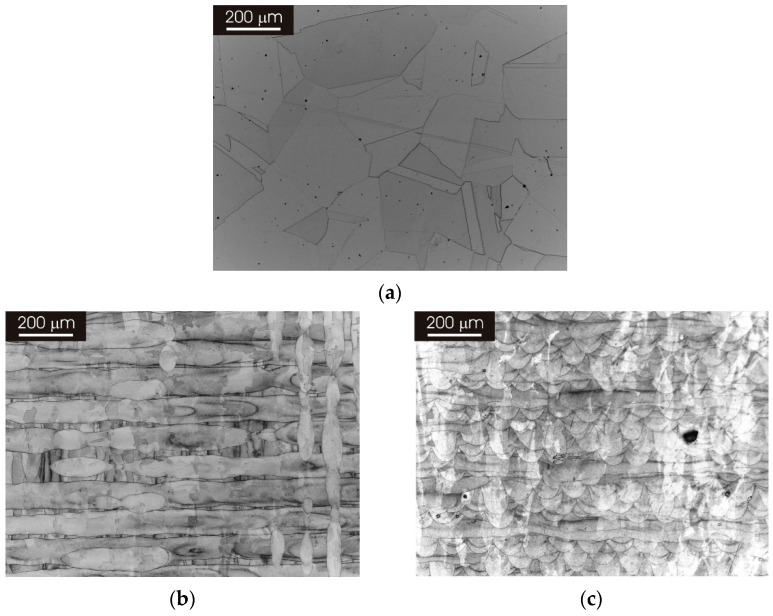
Optical micrographs of 316L stainless steel. (**a**) Wrought condition; (**b**) after 3D printing by L-PBF in the XY direction; (**c**) after 3D printing by L-PBF in the XZ direction.

**Figure 2 materials-15-06768-f002:**
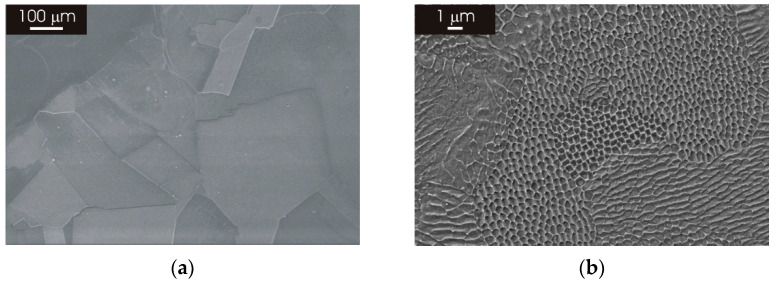
SEM micrographs of 316L stainless steel in the wrought condition. (**a**) Wrought condition; (**b**) after 3D printing by L-PBF in the XY direction.

**Figure 3 materials-15-06768-f003:**
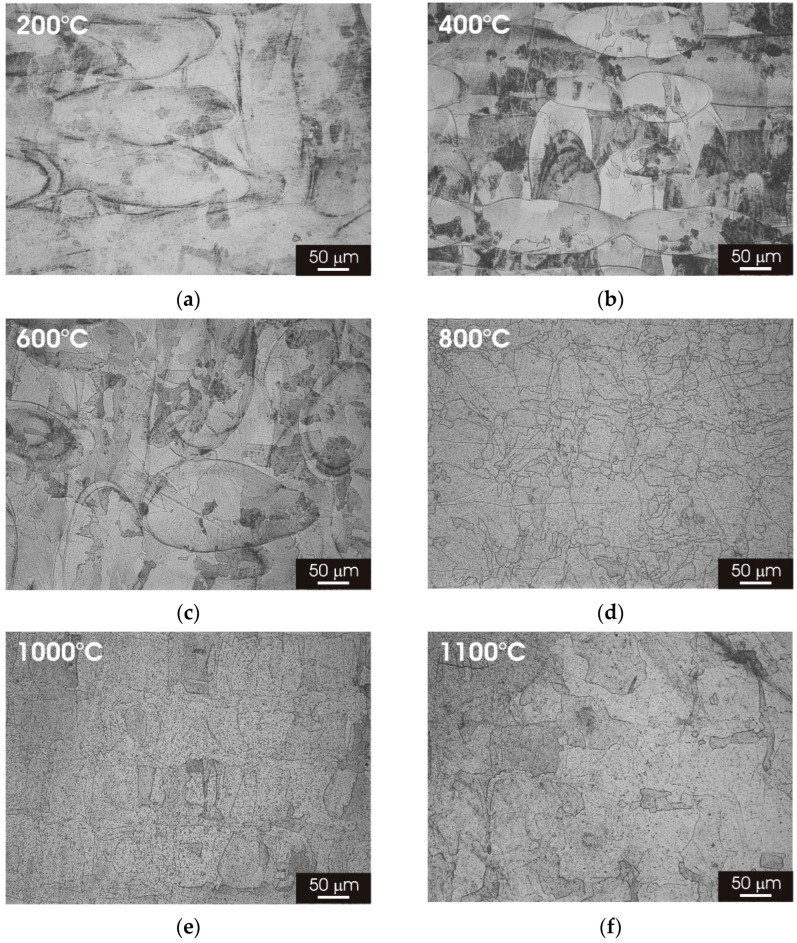
Optical micrographs of L-PBF-XY 316L stainless steel after 2 h thermal treatment. (**a**) 200 °C; (**b**) 400 °C; (**c**) 600 °C; (**d**) 800 °C; (**e**) 1000 °C; (**f**) 1100 °C.

**Figure 4 materials-15-06768-f004:**
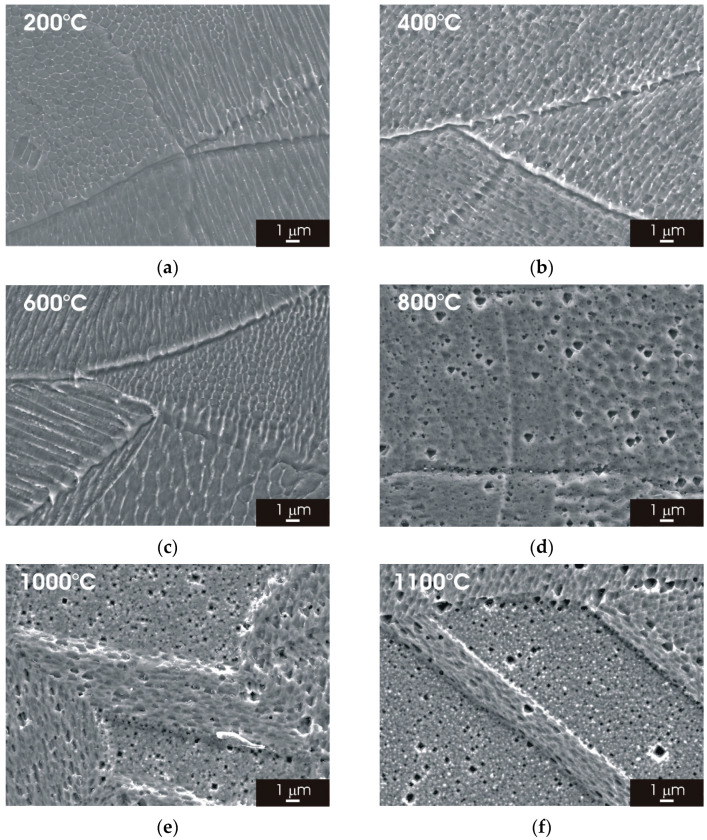
SEM micrographs of L-PBF-XY 316L stainless steel after 2 h thermal treatment. (**a**) 200 °C; (**b**) 400 °C; (**c**) 600 °C; (**d**) 800 °C; (**e**) 1000 °C; (**f**) 1100 °C.

**Figure 5 materials-15-06768-f005:**
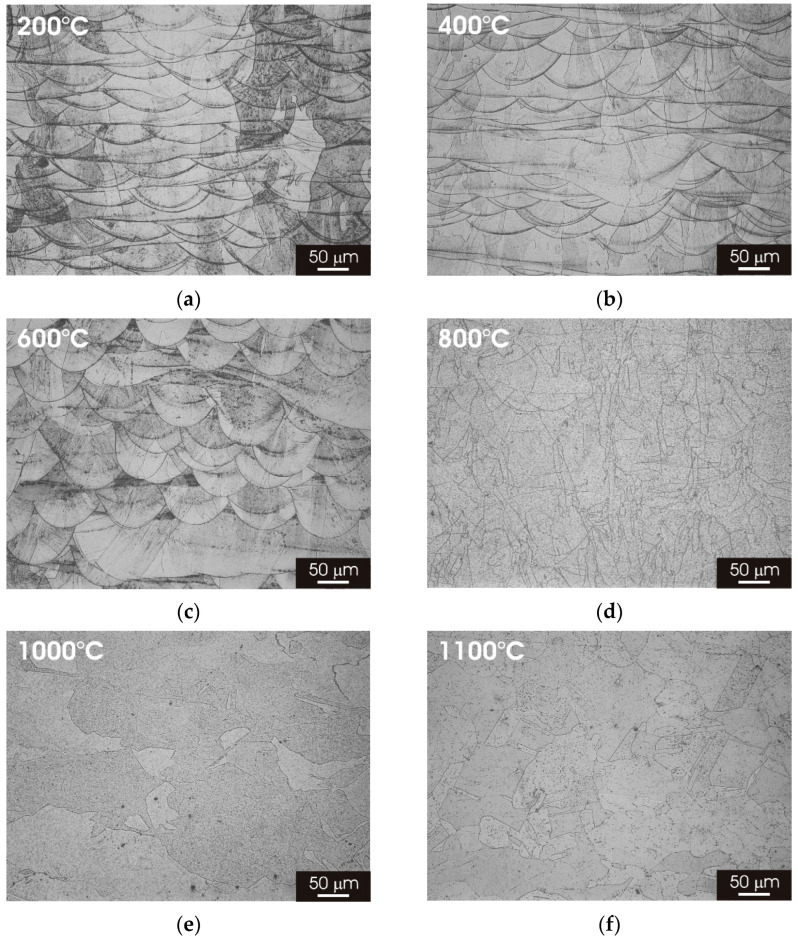
Optical micrographs of L-PBF-XZ 316L stainless steel after 2 h thermal treatment. (**a**) 200 °C; (**b**) 400 °C; (**c**) 600 °C; (**d**) 800 °C; (**e**) 1000 °C; (**f**) 1100 °C.

**Figure 6 materials-15-06768-f006:**
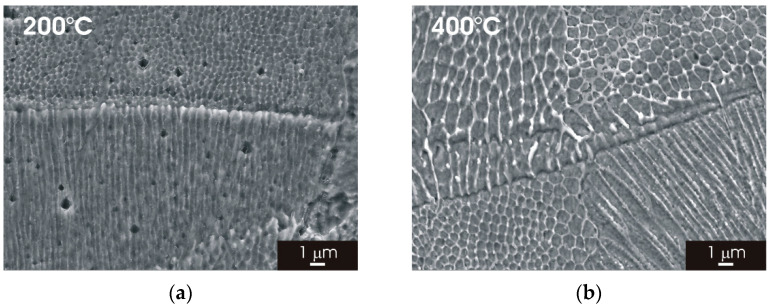
SEM micrographs of L-PBF-XZ 316L stainless steel after 2 h thermal treatment. (**a**) 200 °C; (**b**) 400 °C; (**c**) 600 °C; (**d**) 800 °C; (**e**) 1000 °C; (**f**) 1100 °C.

**Figure 7 materials-15-06768-f007:**
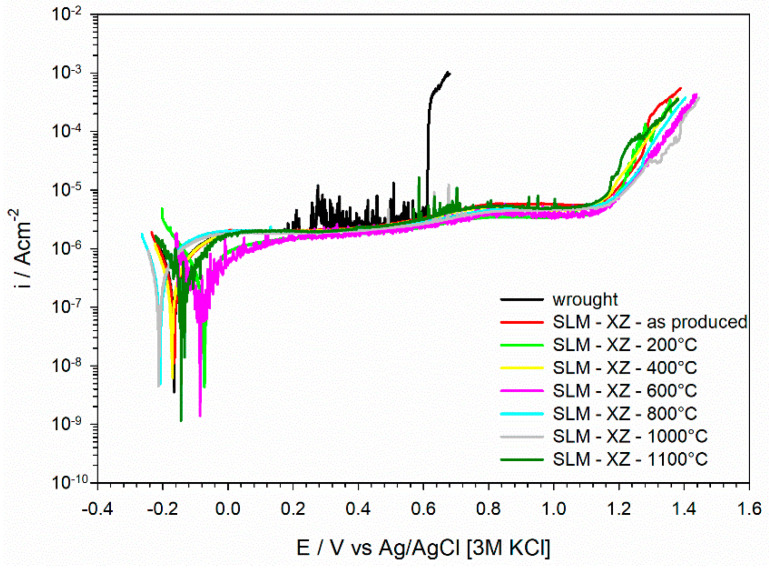
Potentiodynamic polarization curves in neutral 3.5% NaCl solution.

**Figure 8 materials-15-06768-f008:**
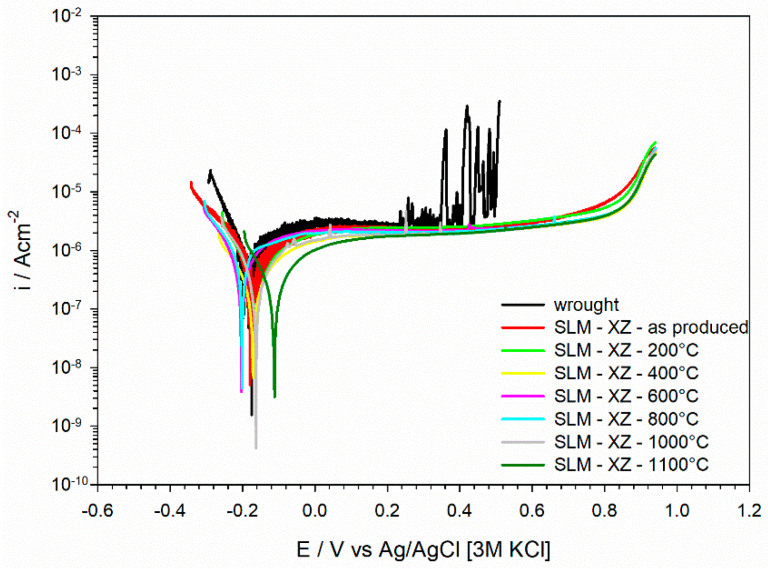
Potentiodynamic polarization curves in 3.5% NaCl solution with pH 1.8.

**Figure 9 materials-15-06768-f009:**
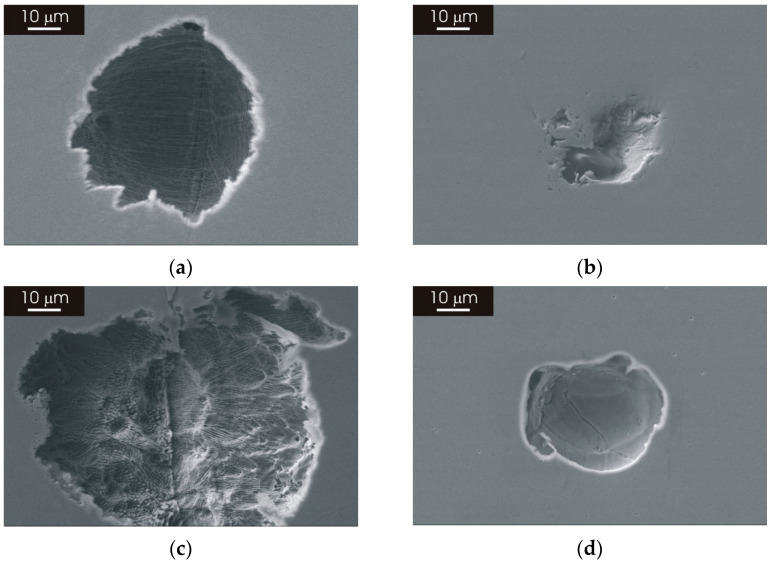
SEM micrographs of heat-treated L-PBF-XY 316L stainless steel after galvanostatic polarization at 1 mA/cm^−2^ for 1 h in 3.5 NaCl solution. (**a**) Heat treated at 400 °C for 2 h, galvanostatic polarization in neutral solution; (**b**) heat treated at 1100 °C for 2 h, galvanostatic polarization in neutral solution; (**c**) heat treated at 400 °C for 2 h, galvanostatic polarization in solution with pH 1.8; (**d**) heat treated at 1100 °C for 2 h, galvanostatic polarization in solution with pH 1.8.

**Figure 10 materials-15-06768-f010:**
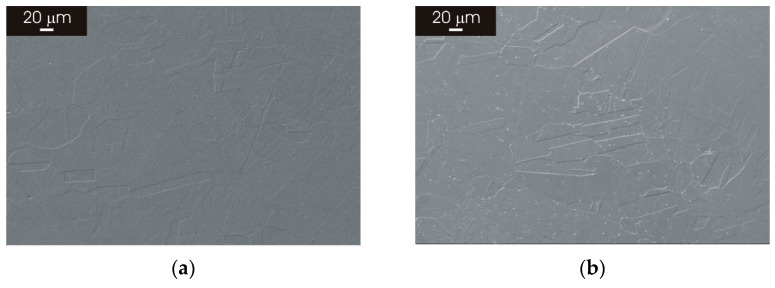
SEM micrographs of L-PBF-XZ 316L stainless steel heat treated at 1100 °C (**a**) heat treatment for 8 h; (**b**) heat treatment for 24 h.

**Figure 11 materials-15-06768-f011:**
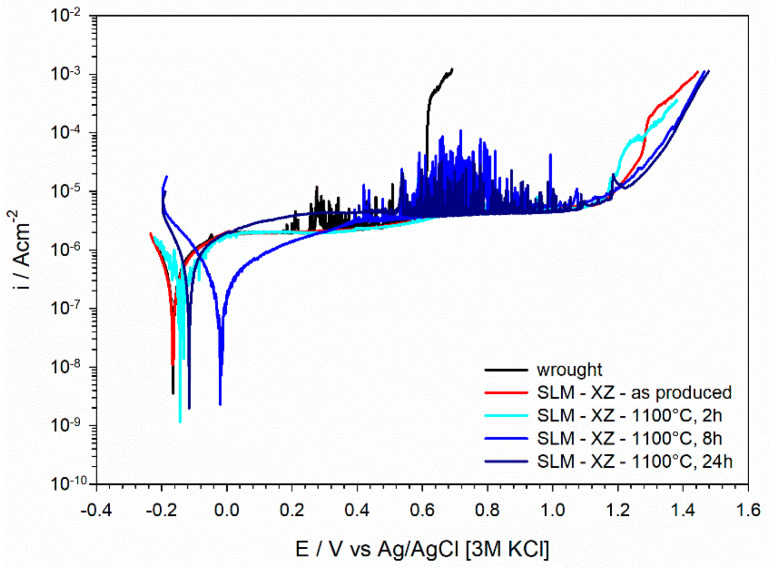
Potentiodynamic polarization curves in 3.5% NaCl solution.

**Figure 12 materials-15-06768-f012:**
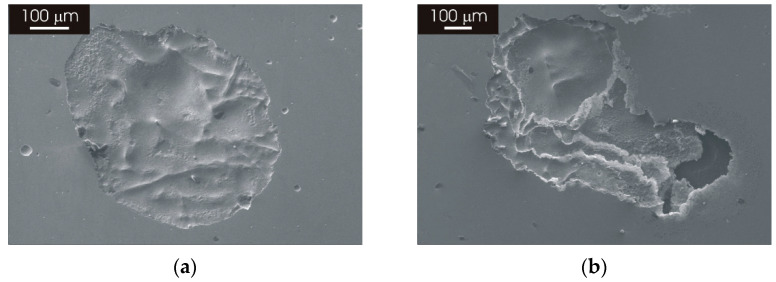
SEM micrographs of L-PBF-XY 316L stainless steel after galvanostatic polarization at 1 mA/cm^−2^ for 1 h in 3.5 NaCl solution. (**a**) Heat treated at 1100 °C for 8 h; (**b**) heat treated at 1100 °C for 24 h.

**Table 1 materials-15-06768-t001:** Chemical composition of L-PBF and wrought AISI 316L stainless steel.

	Cr [wt%]	Ni [wt%]	Mo [wt%]	Mn [wt%]	Si [wt%]	C [wt%]	S [wt%]	N [wt%]	0 [ppm]
wrought	17.4	12.1	2.1	1.6	0.3	0.020	0.002	0.036	48
L-PBF	17.6	12.8	2.3	1.0	0.7	0.023	0.004	0.083	238

## Data Availability

Not applicable.

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
