# Peer review of "Effect of Thermal Treatment on Corrosion Behavior of AISI 316L Stainless Steel Manufactured by Laser Powder Bed Fusion"

_materials, 2022, doi:10.3390/ma15196768_

Round 1

Reviewer 1 Report

The effect of post-heat treatment on the corrosion behaivor of AM 316L SS have been frequently reported. Therefore, the main difference from previous studies and innovation point must be declared. Moreover, previous studies reported that the post-annealing shows great effect on the corrosion behaivor of AM 316L SS, while a opposite conclusion is found in this work, why? 

Reviewer 2 Report

The article generally shows the micrographs and corrosion morphologies, but it fails to explain the correlation between microstructure and electrochemical corrosion. It also does not explain the following details: OCP, rate of corrosion (mm/year or mpy), corrosion kinetics, and the physical phenomenon behind it.

1.      What is the importance of preparing the alloy in two different scan patterns? Which needs to explain in the introduction.

2.      Why has the electrochemical test been conducted with adjusted pH of NaCl solution? What is the importance of using H2SO4 along with NaCl solution?

3.      It is recommended to represent the value of open circuit potential in the article.

4.      While defining potential swept, why does the author mention more cathodic direction instead of anodic direction? Conventionally it is ±0.5V or ±V of OCP.

5.      In session 3.1, it is mentioned as “austenitic structure,” but how the author confirms the existence of austenitic structure?

6.      It is recommended to combine Figures 1 and 2 and also include the SEM micrograph of alloy prepared in the XZ direction.

7.      The corrosion rate of the studied alloy is not mentioned anywhere in the article. It is recommended to mention the corrosion rate.

8.      The article is focused on micrographs and corrosion morphology. Still, it failed to explain the correlation between the microstructure and corrosion rate, which should be the ultimate aim of the article.

9.      It is mentioned that “oxide film” in line no. 376. How does the author confirm the presence of oxide film in the corroded alloy, and what is the oxide in it?

10.  Even in corrosion morphology, the article explains the features of the morphology; it doesn’t reveal the physical phenomenon or chemical reaction behind it.

11.  The conclusion gives the general statements. It is recommended to rewrite the conclusions  by specific the study findings.

Reviewer 3 Report

The authors studied the effect of annealing over a wide temperature range on the evolution of the structure of the L-PBF processed Cr-Ni-Mo-steel and then correlated the observed changes with corrosion resistance. I have two opinions about this article. Points Detracting: the description and discussion of the results are very simple; a deeper study of the problem suggests itself. Points In Favor: the authors demonstrated a good result, namely, the lack of influence of structural changes of the L-PBF processed Cr-Ni-Mo-steel on its high corrosion resistance, which expands the applicability of this material. With this in mind, and given that the topic of the work is very relevant, I will still recommend the article for publication, however, before final acceptance, a number of problems need to be addressed.

1) I would like to see more detailed structural studies, for example, by the SEM method. The possibilities of OM are clearly not sufficient. A deeper description of the results and their analysis are also recommended.

2) Lines 118-121. It's out of place here.

3) Line 161. Seems like it should be 2.3, not 2.2.

4) Line 224. Seems like it should be Figure 4, not Figure 3.

5) Unfortunately, the Supplementary Materials is not available, so I was not able to see the Figures S1 and S2.

6) Parts of Figure 2 need to be done in the same style: signatures at the bottom of the figure should be removed; such information is not necessary. Leave only the scale bar, for example, as in Figure 1. Same for other similar figures.

Author Response

Please see the attachment:

Round 2

Reviewer 1 Report

This paper can be published in present formation.

Reviewer 2 Report

The manuscript has been significantly improved by the authors.
In its current form, the paper can be accepted.

Reviewer 3 Report

The article has been improved after revision. Now I recommend to accept it in present form.